# Relation between Strawberry Fruit Redness and Bioactivity: Deciphering the Role of Anthocyanins as Health Promoting Compounds

**DOI:** 10.3390/foods13010110

**Published:** 2023-12-28

**Authors:** Elsa Martínez-Ferri, Tamara Yuliet Forbes-Hernandez, Lucía Cervantes, Carmen Soria, Maurizio Battino, María Teresa Ariza

**Affiliations:** 1Instituto Andaluz de Investigación y Formación Agraria y Pesquera (IFAPA), Consejería de Agricultura, Pesca y Desarrollo Rural, Junta de Andalucía, IFAPA de Churriana, Cortijo de la Cruz s/n, Churriana, 29140 Málaga, Spain; elsa.martinez@juntadeandalucia.es (E.M.-F.); lucia.cervantes@juntadeandalucia.es (L.C.); maria.soria@juntadeandalucia.es (C.S.); 2Unidad Asociada de I+D+i IFAPA-CSIC Biotecnología y Mejora en Fresa, 29140 Málaga, Spain; 3Department of Physiology, Institute of Nutrition and Food Technology José Mataix Verdú, Biomedical Research Center, University of Granada, Avda. Del Conocimiento s.n. Armilla, 18100 Granada, Spain; 4Dipartimento di Scienze Cliniche Specialistiche ed Odontostomatologiche (DISCO)-Sez. Biochimica, Facoltà di Medicina, Università Politecnica delle Marche, 60131 Ancona, Italy; m.a.battino@univpm.it

**Keywords:** achenes, antioxidants markers, enzymatic activity, ROS, white fruit

## Abstract

The red colour of most berries is often associated to fruit healthiness, since it has been linked to enrichment in anthocyanins (polyphenol with antioxidative properties). However, recent studies suggest that anthocyanins could not be the major contributors to bioactivity leading to uncertainty about their role as important molecules in the generation of health-promoting properties. To shed light on this issue, spectrophotometric and HPLC techniques were used for characterizing the content of phenolic compounds, including anthocyanins, in fruits of red (*Fragaria x ananassa*, cv. Fortuna) and white strawberry (*Fragaria vesca* spp. XXVIII) species (distinguishing receptacle from achene). In addition, the effect of these extracts on the reduction of intracellular ROS was tested, as well as on the activity of antioxidant enzymes and the quantification of cell oxidation markers. The results showed that white receptacle extracts (deprived of anthocyanins) were able to protect cells from oxidative damage to a greater extent than red fruits. This could be due per se to their high antioxidant capacity, greater than that shown in red fruits, or to the ability of antioxidants to modulate the activity of antioxidant enzymes, thus questioning the positive effect of anthocyanins on the wholesomeness of strawberry fruits. The results shed light on the relevance of anthocyanins in the prevention of health-associated oxidative damage.

## 1. Introduction

Red berry fruits—widely consumed worldwide—are considered healthy fruits, since they contain diverse bioactive compounds linked to multiple health benefits [1,2,3]. These health benefits are commonly associated to the presence of antioxidants, mainly polyphenols [4,5,6,7].

Among red fruits, strawberries are one of the most widely accepted by consumers. In addition, many studies have shown that strawberries have beneficial substances for health, both in the receptacle and in the achenes (i.e., real dry fruits [8,9]). Although achenes represent a very low percentage of the fruit weight, they contain a large amount of antioxidants, and are important contributors to the antioxidant capacity of whole fruit, being responsible for one of the health promoting properties attributed to strawberry fruit [10].

Among the antioxidants beneficial to health, phenolic compounds are one of the most important, because of their antioxidant and anti-inflammatory action, and because they directly and indirectly present antimicrobial, antiallergy, and anti-hypertensive properties [11,12,13]. In addition, many studies have shown that fruit phenolic compounds counteract cellular oxidative stress by decreasing reactive oxygen species (ROS) and through modulation of some physiological enzymes and receptors activities, thus preventing oxidative stress-related diseases [14].

The main class of polyphenols in strawberries are flavonoids, mainly anthocyanins, a class of water-soluble flavonoids, responsible for the intense red and blue colours of numerous fruits and flowers [15,16] and, thus, for the bright red colour of fresh strawberries [17]. Different studies have shown that strawberries have a simple anthocyanin profile, with pelargonidin 3-glucoside as the predominant pigment, followed by cyanidin 3-glucoside [10,18].

Anthocyanins have antioxidant [19,20], anticarcinogenic [21,22], vasoprotective [23], and anti-inflammatory [24] properties. They have also been described to improve vision [25] and memory [26], and have anti-obesity effects [27]. In fact, since many berries exhibit health-promoting properties and anthocyanins are the major compounds of these fruits, many authors have suggested that anthocyanins are responsible for the health-promoting properties described above [28]. For example, several studies have reported the health potential of anthocyanins. Thus, some studies have shown that an anthocyanin-rich strawberry extract protected human dermal fibroblasts against oxidative stress [19] and others have shown that anthocyanin extracts inhibited the proliferation of melanoma cells in mice [29], implying a main role of anthocyanins in human health. However, other studies have suggested that anthocyanins could not play a major role in these health benefits, as they are not a major contributor to antioxidant capacity [30,31]. However, this is not clear, as anthocyanins are often bound to other flavonoids, enhancing their biological effects and making it difficult to decipher the contribution of these components [32].

Moreover, although most of the health properties of strawberries seem to come from molecules with antioxidant action, such as polyphenols, recent research has shown that the biological and functional activities of polyphenols are related not only to antioxidant capacity but also to the modulation of many cellular pathways involved in metabolism, survival, proliferation, and antioxidant defences [14], such as antioxidant enzymes (catalase, superoxide dismutase) or antioxidant molecules (glutathione, among others).

Therefore, the objectives of the present study were to evaluate to what extent anthocyanins are involved in ameliorating cell oxidative stress, modulation of antioxidative enzymes, and/or of antioxidant molecular markers in vitro, and to determine the role of anthocyanins in the health effect attributed to strawberries.

## 2. Materials and Methods

### 2.1. Plant Material and Experimental Design

Three replications of ~250 g of red ripe fruits of strawberry (*Fragaria x ananassa* Duch.) cv. ‘Fortuna’ and of white ripe fruits of wild strawberry (*Fragaria vesca* spp. XXVIII) were picked up from strawberry germplasm bank located at Ifapa Center in Málaga, Spain. The achenes were then removed by hand from the fruits and the achene-free receptacle was homogenized in a Multiquick MR 6500 blender (Braun, Kronberg, Germany). Both the receptacle puree and achenes were stored at −20 °C until analysis.

### 2.2. Determination of Antioxidant Groups

The main antioxidant groups, the antioxidant capacity, and the individual phenolic compounds in ripe fruits of each genotype were determined. For this purpose, hydromethanolic extracts obtained from ~1 g receptacle puree or 0.2 g achenes added to 10 mL of the extraction solution consisting of methanol/MilliQ water/HCl (80:19.9:0.1 *v*/*v*) were homogenized for 2 h at room temperature [7]. The extracts were filtered and centrifuged at 10,000 rpm for 10 min at 4 °C, and the supernatant was stored at −20 °C until spectrophotometric and HPLC analyses.

The phenolic compounds and antioxidant capacity of the hydromethanolic extracts of the receptacle and achenes of each genotype were determined in triplicate by UV–Vis spectrophotometry, using a Shimadzu PharmaSpec UV-1700 instrument (Kyoto, Japan). The total phenolic content (TPC), total flavonoid content (TFC), total anthocyanin content (TAC), and total tannin content (TTC), as well as the antioxidant capacity (AC) assessed by Trolox Equivalent Antioxidant Capacity assay (TEAC), were determined according to [10]. Briefly, for the determination of TPC, 100 µL of samples or gallic acid standard solution were mixed with 500 µL Folin–Ciocalteu’s reagent (10%) and incubated for 3 min, then 400 µL of sodium acetate solution (7 M) was added. The absorbance was measured after 2 h of incubation, at 760 nm in a UV-Vis spectrophotometer. For TFC determination, 250 µL of samples or catechin standard solution were mixed with 1.25 mL of MilliQ water and 75 µL of 5% NaNO_2_ solution. After 6 min, 150 µL of 10% AlCl_3_ 6H_2_O solution was added and the mixture was incubated for 5 min, then 500 µL of 1 M NaOH was added. The volume was brought to 2.5 mL and the absorbance was measured at 510 nm. TAC was measured using the differential pH method (0.025 M potassium chloride (KCl) buffer, pH 1.0 (buffer 1) and 0.4 M sodium acetate (CH_3_CO_2_Na) buffer, pH 4.5 (buffer 2) and measuring absorbance at 510 and 700 nm). For TTC, 200 µL of hydromethanolic extract were centrifuged with 1800 µL of N-N dimethylformamide. Then, 500 µL of supernatant was mixed with 2.5 mL of H_2_O, 0.5 of ammonia solution, and 0.5 mL of ferric citrate solution. The absorbance was then measured at 525 nm. The AC assay uses ABTS molecules. The absorption maximum of ABTS is at 734 nm. The addition of antioxidant compounds reduces ABTS to its colourless form, so the degree of decolorization (measured as percentage inhibition of ABTS) was determined as a function of concentration and calculated in relation to the reactivity of Trolox.

Results of TPC, TFC, TAC, TTC, and AC were expressed as milligrams of gallic acid equivalents (GAE), catechin equivalents (CAE), pelargonidin-3-glucoside equivalents (PE), tannic acid equivalents (TAE), and micromols of Trolox equivalents (TE) per 100 g of fresh weight (FW), respectively.

### 2.3. Individual Phenolic Compounds

Individual phenolic compounds were analysed on an Agilent 1200 HPLC system (Agilent Technologies, Palo Alto, CA, USA) operated by a Windows based ChemStation software, A.01.02. The HPLC equipment consisted of a quaternary pump, degasser, and auto sampler, and the system was used with a diode array detector (DAD). The column used was a Zorbax Eclipse Plus C-18 LC column (3.00 mm × 150 mm, 3.5 µm) furnished with a guard column (4.6 mm × 12.5 mm, 5 µm), both from Agilent. All samples were filtered through a 0.45 µm filter (type Captiva Econofltr Nylon 25 mm, Agilent Technologies) prior to HPLC analysis.

The reagents and HPLC conditions were previously described by [10] and the individual antioxidants (gallic acid, protocatechuic acid, procyanidin B1 and B2, 4-hydroxibenzoic acid, catechin, vanillic acid, caffeic acid, syringic acid, epicatechin, p-coumaric acid, trans-ferulic acid, ellagic acid, trans-cinnamic acid, cyanidin-3-glucoside, pelargonidin-3-glucoside, pelargonidin-3-rutinoside, myricetin, quercetin, and kaempferol) were evaluated according to the procedure described by [10].

### 2.4. Bioactivity Assay of Strawberry Extracts

The effect of strawberry receptacle and achenes extracts on cells was evaluated by analysing intracellular ROS production in a culture of human hepatocellular carcinoma (HepG2) cells.

HepG2 cells were purchased from the American Type Culture Collection (ATCC^®^ CL-173TM) and were grown in Dulbecco’s modified Eagle’s medium supplemented with 10% foetal bovine serum, 100 IU/mL penicillin, and 100 µg/mL streptomycin until 80–90% confluence was reached, at which time they were sub-cultured. Cells were maintained in an HeraCell CO_2_ incubator at 37 °C with 5% CO_2_.

The time of application and the dose used of each strawberry extract were selected taking into account a cell viability test (MTT assay). However, the time of application and dose used of the stressor agent AAPH (2,2’-Azobis (2-amidinopropane) dihydrochloride; CAS number: 2997-92-4) was previously determined [33].

To assess cell viability of strawberry and achene extracts, cells were seeded into 96-well plates at a density of 5 × 10^3^ cells/well, allowed to adhere for 16 h, then treated with different concentrations of strawberry receptacle (0, 5, 10, 25, 50, 100, 250, 500, 1000, 2500, 5000, 7500, and 10,000 µg/mL) and different concentrations of achenes extracts (0, 0.25, 0.5, 1, 5, 10, 25, 50, 100, 250, and 500 µg/mL) for 24, 48, and 72 h. After incubation, 30 µL of RPMI medium containing 2 mg/mL of MTT (3-(4,5-Dimethylthiazol-2-Yl)-2,5-Diphenyltetrazolium Bromide) was added to each well. The cells were then incubated for 2 h at 37 °C in a 5% CO_2_ incubator. The MTT solution was then discarded and 100 µL of dimethyl sulfoxide was added to each well to dissolve the formazan crystal. The level of coloured formazan derivative was analysed on a microplate reader (Thermo Scientific Multiskan EX, Monza, Italy) at a wavelength of 590 nm. The percentage of cell viability was calculated according to the following equation:% cell viability = (treated cells Abs/control cells Abs) × 100
where Abs is absorbance.

Data are presented as the mean value of three independent analyses ± standard error (SE).

For each strawberry extract, as well as for the stressor, the highest concentrations, which ensured a cell viability higher than 90%, were selected for further analysis.

Determination of intracellular ROS levels was performed using CellROX^®^ Orange reagent (Invitrogen, Life Technologies, Milan, Italy) according to the manufacturer’s instructions, following the procedure previously reported by [33]. Briefly, cells were seeded in 6-well plates at a density of 1.5 × 10^5^ cells/well and treated for 24 h with AAPH or selected concentrations of strawberry extracts. After that, half batches of cells treated with the strawberry extracts were post-incubated in presence of 2.5 mM of AAPH for 24 h. Each treatment was performed in three replicates and the results were reported as a fold increase compared to the control.

### 2.5. Determination of Antioxidant Enzyme Activities and Markers of Oxidation

In this study, representative antioxidant enzymes such as superoxide dismutase (SOD), catalase (CAT), glutathione reductase (GR), and glutathione-S-transferase (GT), together with the total amount of glutathione (GSH) and another oxidation indicator, such as thiobarbituric acid-reactive substance (T-BARS) were analysed to assess oxidative stress status in cells. Thus, HepG2 cells were incubated with RIPA buffer on ice for 5 min and the lysate was analysed as previously reported by [34]. Briefly, SOD activity was determined from the inhibition of NADH-phenazine methosulfate-nitroblue tetrazolium (NBT) formazone formation by SOD. CAT was tested following hydrogen peroxide decomposition. One unit of CAT was defined as the amount of enzyme that decomposed 1 mmol H_2_O_2_ per minute per mg protein. GR activity was determined by checking glutathione-dependent NADPH oxidation at 340 nm. The reaction was initiated by adding GSSG and the oxidation rate was calculated using the extinction coefficient of NADPH. The activity of GT was evaluated by measuring the conjugation of 1-chloro-2,4-dinitrobenzene (CDNB) to reduced glutathione yielding a dinitrophenyl thioether. One unit of GST activity is defined as the amount of enzyme producing 1 mmol of CDNB-GSH conjugate per minute per mg of protein. For GSH determination, 1 mL of the sample extract was treated with 4.0 mL of metaphosphoric acid solution. After centrifugation, 2 mL of the supernatant was added to 0.2 mL of 0.4 M Na_2_HPO_4_ and 1 mL of DTNB reagent. The absorbance was read at 412 nm within 2 min. Finally, T-bars was evaluated by adding the lysate to 2-thiobarbituric acid (TBA). After heating at 95 °C for 30 min, the mixture was cooled in an ice bath and centrifuged at 10,000× *g* for 10 min. The absorbance of the supernatant was read at 532 nm.

Each determination was carried out in three replicates and the results were expressed as a fold increase with respect to the control.

### 2.6. Statistical Analysis

Data were analysed by one-way analysis of variance (ANOVA) using Statistix software 9.0 (Analytical Software, Tallahassee, FL, USA) and differences, among means, were located using Tukey’s honest significant difference (HSD) test. Homogeneity of variances and normality were tested using the Levene test and the Shapiro–Wilk test, respectively. The R program [35], together with the FactoMineR and factoextra packages were used to apply principal component analysis (PCA) to examine relationships between the antioxidant compounds in strawberry extracts.

## 3. Results and Discussion

Spectrophotometric and HPLC assays were performed to determine the total antioxidant content of the main polyphenolic groups and the profile of individual phenolic compounds in both the receptacle and achenes extract of red and white strawberry fruits. In addition, bioactivity assays were performed to evaluate the protective effects of these extracts against oxidative damage in HepG2 cells, including their action on antioxidant enzyme activity and antioxidant marker levels.

### 3.1. Polyphenolic Composition

#### 3.1.1. Main Polyphenol Groups and Antioxidant Capacity

Table 1 shows the amount of total phenolic content (TPC), total flavonoids content (TFC), total anthocyanin content (TAC), and total tannin content (TTC), as well as the antioxidant capacity (AC) in red and white fruits, in both achene and receptacle. Based on these data, the phytochemical characterization was similar to that previously reported for many strawberry genotypes [36,37,38].

White receptacles showed more TFC and AC than the red ones; however, red achenes showed higher TFC and TAC than white ones. Although the contribution of achenes to total fruit weight is approximately 0.7% [10], the percentage attributed to achenes, in relation to the different phenolic groups, is surprisingly high, as previously described by [10]. Thus, achenes were responsible for 17.66%, 11.18%, 21.16%, and 12.71% of TPC, TFC, TTC, and AC respectively in red fruits, and 19.67%, 7.14%, 23.05%, and 10.04% of TPC, TFC, TTC, and AC respectively in white fruits. Regarding TAC, as expected, they were not detected in the receptacle or achenes of white fruit, and in red fruits most of them (99.56%) were found in the receptacle.

Many studies have underlined the major role played by anthocyanins in the health promoting properties of berries [18,28,29], such as reducing the risk of cardiovascular disease [39] or decreasing the risk of hypertension [40], attenuating endothelial inflammation and oxidative stress, improving the lipid profile and modifying vascular glycocalyx, and reducing type-2 diabetes [41] by decreasing blood glucose and glycated haemoglobin levels, reducing certain types of cancers [42] by inhibiting apoptosis, cell growth and differentiation, inflammatory responses, and oxidative stress [43], or preventing cognitive disorders [44], among others.

Since many health effects attributed to red fruits seem to be promoted by the antioxidant capacity (AC; i.e., the ability of compounds to neutralize and scavenge radicals), a large contribution of anthocyanins to AC is to be expected; however, white fruits (lacking anthocyanins) showed higher AC than red fruits (Table 1). In this sense, a higher AC did not correspond to high values of either TAC, TPC, or TTC, and therefore other compounds, such as vitamins and antioxidant enzymes, among others, could be involved in AC, as suggested [36,45]. This result indicates that the previously described effects of anthocyanins on health could be mostly related to their ability to modulate different molecular pathways [46,47,48,49] rather than their contribution to AC.

#### 3.1.2. Individual Phenolic Compounds

The individual phenolic compounds in the receptacle and achenes of red and white strawberry fruits are displayed in Table 2.

In general, the red receptacles showed higher amounts of the four main groups of individual phenols (non-flavonoid type, which include phenolic acids and hydrolysable tannins (ellagitannins) and flavonoid type, comprising anthocyanins, flavonols, and flavan-3-ols (including proanthocyanidins)) than the white ones. However, the latter showed more catechin that the reds, in consonance with the higher amount of flavonoids found in the white receptacle (Table 1), which could be one of the reasons for this higher amount of flavonoids in white receptacles. This fact could be relevant since catechin is a flavanol that exhibits many beneficial properties, such as anticancer, anti-obesity, antidiabetic, anticardiovascular, anti-infectious, hepato-protective, and neuroprotective effects [50]; therefore, highly beneficial health properties could be attributed to the white receptacle, and catechin could be one of those responsible for the high AC shown in the white receptacle (Table 1), since clinical studies have shown the beneficial effects of catechin due its antioxidant action [51].

In the case of the red fruits, although pelargonidins and cyanidins are predominant in strawberry red fruits, they are scarcely present in cv. Fortuna, in agreement with [30]. Furthermore, it is surprising that neither quercetin nor kaempferol, which are among the most abundant flavonols in red strawberry fruits [52], were quantified.

In general, achenes showed higher quantity but lower diversity of individual phenols than receptacles. In particular, ellagic acid was the majority in achenes. This fact is very important since it is one of the most described antioxidants with high health properties [53] and could be one of the compounds by which strawberries as described as healthy. Regarding fruit type, the amount of ellagic acid in both receptacle and red achenes was higher than in the white ones (three-fold and two-fold respectively).

As expected, anthocyanins were not measured in the white fruits, neither in the receptacle nor in achenes. However, although the achenes of red fruits are not reddish, their anthocyanins content is remarkable, in agreement with [10].

It is noteworthy that white achenes showed higher values of flavanols (Table 2), and specifically exhibited nearly 13-fold more myricetin than red achenes. Since myricetin has been associated with anticancer, antidiabetic, anti-obesity, cardiovascular protection, osteoporosis protection, anti-inflammatory, and hepato-protective effects [54], white achenes could be considered a very interesting food matrix.

Analysing the relationship between the data of the total antioxidant content of the main polyphenolic groups and those of the individual phenolic compounds by principal component analysis, it was observed that the first two principal components (PCA1 and PCA2) accumulated a fairly high proportion of variance, around 90%; specifically, PCA1 explained 65.50% of the total variance and PCA2 explained 24.12% (Figure 1).

In this regard, AC appears to be related to a greater extent to TTC and TPC, followed by TFC. However, TAC does not seem to be related to AC, as mentioned above in Table 1. Regarding individual compounds, it seems that catechin seems to contribute less to the antioxidant capacity, but cyanidin-3-glucoside (which is a minority anthocyanin in ‘Fortuna’) seem to contribute more, indicating that although anthocyanins (TAC) do not seem to contribute to AC, some particular anthocyanins might, but being a minority, it would not be reflected in the overall behaviour.

### 3.2. Bioactivity Assay

To evaluate the potential cytotoxicity of strawberry receptacle and achene extracts, cell viability (MTT assay) was assessed in HepG2 cells by exposure with different dilutions of red or white fruit receptacles or achenes extracts after 24, 48, and 72 h (Figure 2).

Reduced cell viability caused by more concentrated extracts or longer exposure time has been displayed in other studies. In this regard, [55] showed that high concentrations of bioactive contained in red fruits compounds inhibited the cell cycle in the G2/M phase and caused cell death, exerting a negative impact on cell viability. In addition, these authors demonstrated that high concentrations of phenolic extracts of berries induced apoptosis due to activation of caspases.

After evaluation of potential cytotoxicity, two concentrations of each extract were chosen that guaranteed cell viabilities above 90%, selecting 10 and 25 µg/mL for receptacles and 0.25 and 0.5 µg/mL for achenes. For the stressor agent (AAPH), the concentration was selected based on our previous studies (2.5 mM; [33]).

The magnitude of the difference in the concentrations of achenes chosen with respect to those of the receptacles, selected for giving rise to cell viabilities greater than 90%, is remarkable. Thus, the concentrations of receptacles were up to 100-fold higher than those of achenes, suggesting that achenes are a major sink for antioxidants, which is consistent with the quantified amounts shown in Table 1.

To test the efficacy of berry extracts against AAPH-induced stress in HepG2 cells, the measurement of intracellular ROS production is a very useful tool to assess such oxidative stress [56]. The accumulation of ROS can lead to hyperactivation of the inflammatory response and tissue damage, among other reactions [57].

As shown in Figure 3, treatment with AAPH increased the intracellular ROS levels up to 3.7-fold compared to the control (untreated cells).

Application of most of the concentrations of white and red receptacles or achenes on the cells decreased ROS values close to the control baseline, showing no significant difference in their ROS level compared to untreated cells, supporting the position that these concentrations were not cytotoxic, as previously demonstrated by MTT (Figure 2). However, the red receptacle at 25 µg/mL showed significantly higher values with respect to the control, but its ROS values were not as high as when AAPH was applied, which can be considered an elevated concentration but not harmful to the cell. This could be due to the fact that it has been described that a high concentration of antioxidants could lead to pro-oxidation [58], and therefore to an increase in cellular ROS.

However, when receptacle extracts were applied before incubation with AAPH, in most cases they did not cause a reduction in ROS level compared to untreated cells. Only in the case of white receptacles at 25 µg/mL + AAPH was a significant decrease observed with respect to the AAPH group (*p* < 0.05; Figure 3A), reaching ROS values close to those of untreated cells. This suggests that the white receptacle extract must have specific compounds, and that at that concentration (25 µg/mL) they counteract the oxidative effect of AAPH and cause the cells to decrease their ROS values to amounts similar to control cells. In the case of white or red achenes extracts +AAPH, the amount of cellular ROS did not show significant differences with respect to cells to which only AAPH was applied (Figure 3B), showing that these extracts, at the concentrations tested, did not counteract the oxidative effect of AAPH, and therefore these cells would not appear healthy.

Therefore, the effect of the white receptacle counteracting cellular stress (represented by the level of ROS) in AAPH-treated HepG2 cells (Figure 3A) could be explained by the higher amount of AC, which in turn could be attributed to the higher amount of TFC in this food matrix (Table 1), and although it could be the result of an effect of catechins (which are higher in white receptacle; Table 2) that could modulate other cellular elements such as enzymes, among others, it is not due to the antioxidant action of these, since they do not seem to be responsible for the elevation of AC because they show a negative relationship with it (Figure 1).

### 3.3. Antioxidant Enzymes Activity

Cellular redox homeostasis is sustained through a complex antioxidant defence system, which includes non-enzymatic antioxidant compounds and endogenous antioxidant enzymes such as superoxide dismutase (SOD), catalase (CAT), or glutathione-related enzymes, among others [59], these being the most important enzymes in the cellular antioxidant defence system [60]. Among all antioxidant enzymes, CAT converts hydrogen oxide to oxygen and water, and SOD is an antioxidant enzyme that catalyses the conversion of superoxide anion radicals to oxygen and hydrogen peroxide [61]. In addition, glutathione S-transferases (GTs) are involved in the detoxification of xenobiotic compounds and the biosynthesis of key metabolites [62] and glutathione reductase (GR) converts oxidized glutathione (GSSG) to reduced glutathione (GSH), which contributes to maintaining a high GSH/GSSG ratio under various abiotic stresses [63].

Several studies have reported that certain bioactive compounds are able to modulate the activity of antioxidant enzymes, reducing AAPH-induced damage in cells [18,56]. In that sense, both red and white receptacle increased SOD activity with respect to the control baseline (Figure 4), indicating that these extracts could be modulating the enzymatic activity of this enzyme. In the case of achenes, as shown in Figure 5, elevated CAT, SOD, GR, and GT activity, similar or higher than that of the control, were observed when white achene extracts were applied, whereas red achene extracts only increased GR activity while decreasing SOD values, highlighting the different effect of these extracts on the modulation of these enzymes. Furthermore, the observed effects of white achene extracts were dose-dependent for GR and GT activity. Different studies have pointed out this fact, describing the enhancement of SOD and CAT activities in HepG2 cells after treatment with some natural compounds, such as blueberry extract [64] or resveratrol [65].

In all determinations, treatment with AAPH produced a significant reduction (*p* < 0.05) in enzyme activities, indicating that AAPH causes oxidative damage, and that at the concentration tested (2.5 mM) it decreases the activities of antioxidant enzymes, since at this high concentration its addition depletes the enzyme reserve, making it difficult to counteract the damage caused.

However, treatment with AAPH was effectively counteracted by most of the extracts applied, except for GR activity by the 25 µg/mL extract of red receptacle and SOD activity by the 0.25 µg/mL extract of white fruit achenes (Figure 4 and Figure 5, respectively). On the other hand, in the majority of cases, this enhanced activity was not dose-dependent (i.e., the higher concentration was not able to increase enzyme activity and counteract the effect of AAPH to a greater extent).

These strawberry receptacle and achenes extracts, which counteract the harmful effects of AAPH, could act by inducing antioxidant enzymes through the Keap1-Nrf2 system, according to [66,67], which is described as a crucial process for regulating antioxidant enzymes, including SOD, CAT, and glutathione-related enzymes [68].

### 3.4. Oxidation Markers

Regarding antioxidant markers, lipid peroxidation was evaluated to quantify oxidative damage in HepG2 cells treated with AAPH and/or berry extracts. Lipid peroxidation is a free radical-mediated chain reaction, which can be stopped through enzymatic agents or by free radical scavenging by antioxidants [69]. Some diagnostic tests are available for quantification of lipid peroxidation end-products, with the thiobarbituric acid reactive substance (TBARS) assay being the most widely used.

In addition to lipid peroxidation, total reduced glutathione (GSH) level is another important marker of oxidation, since it plays an important role in maintaining the normal reduced state of cells and strongly counteracts the harmful effects of oxidative stress and detoxifying xenobiotics [70].

As shown in Figure 6, most of the receptacle and achenes extracts of both types of fruit maintained GSH levels at the control level, although in the case of the red receptacle extract, GSH decreased with respect to the control, but without reaching the levels of the AAPH-treated cells, which reinforces the idea discussed above (Figure 4) that the red receptacle extract must contain elements that cause partial oxidation of GSH. On the other hand, AAPH treatment showed a decrease in GSH, as the cells are under oxidative stress and GSH is being depleted. Pretreatment of AAPH with berry extracts increased the GSH level, reaching values similar to the control, indicating that berry extracts seem to counteract the oxidative effect of AAPH. Similar results are observed in the case of TBARS. In this sense, receptacle and achenes extracts of both types of fruit did not affect the amount of TBARS compared to untreated cells and, in general, the combined treatment of AAPH with receptacle and achenes extracts decreased the levels of TBARs compared to the AAPH treatment, reaching similar values to the control groups with all extracts and decreasing the lipid peroxidation caused by AAPH.

These results are in agreement with previous studies reporting the ability of different natural bioactive compounds, including strawberry polyphenols, to increase GSH levels and suppress lipid peroxidation in vitro and in vivo in several stressed models [69,71,72,73].

## 4. Conclusions

In conclusion, this study reveals that anthocyanins do not appear to play a key role in the health-promoting properties attributed to strawberries. Thus, white strawberry fruits, lacking anthocyanins, showed a higher antioxidant capacity and decreased cellular oxidative stress to a greater extent than red strawberry fruits, enriched in anthocyanins. Furthermore, this work also highlights that, in general, all strawberry extracts reduced the effect of stressors, decreasing oxidative damage through direct antioxidant action (deter-mined by AC) and/or through the action of antioxidant enzymes, among others.

Our findings point to the importance of deepening the knowledge of the action of strawberry fruit bioactive compounds involved in restoring cellular redox homeostasis, either directly or by modulating cellular antioxidant systems, to better understand the causes underlying the health-promoting properties of strawberries.

## Figures and Tables

**Figure 1 foods-13-00110-f001:**
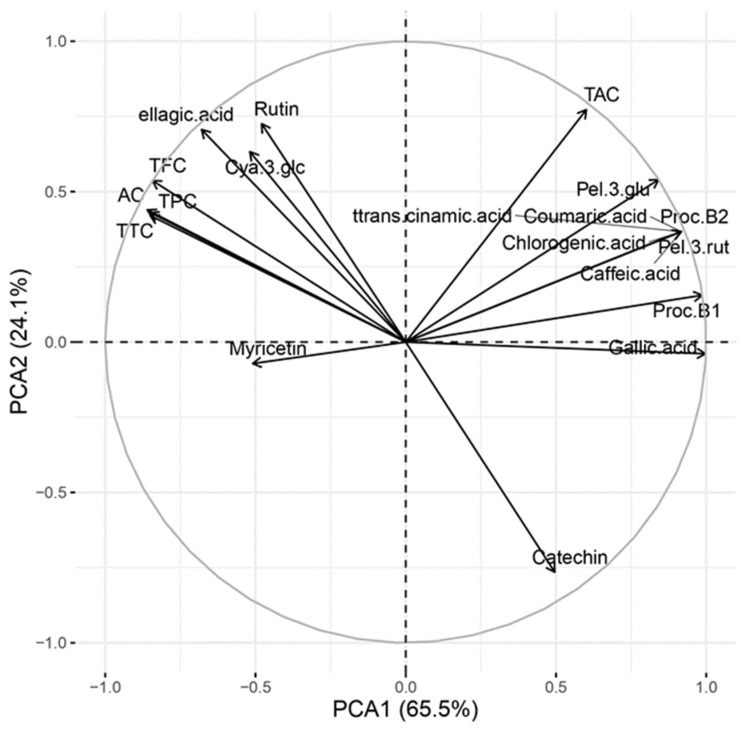
Principal component analysis: Grouping and relationships of antioxidant groups (TPC, TFC, TAC, TTC, and AC) and individual phenolic compounds analysed on red and white extract of achenes and receptacle *vesus* the first two main components (PCA1 and PCA2).

**Figure 2 foods-13-00110-f002:**
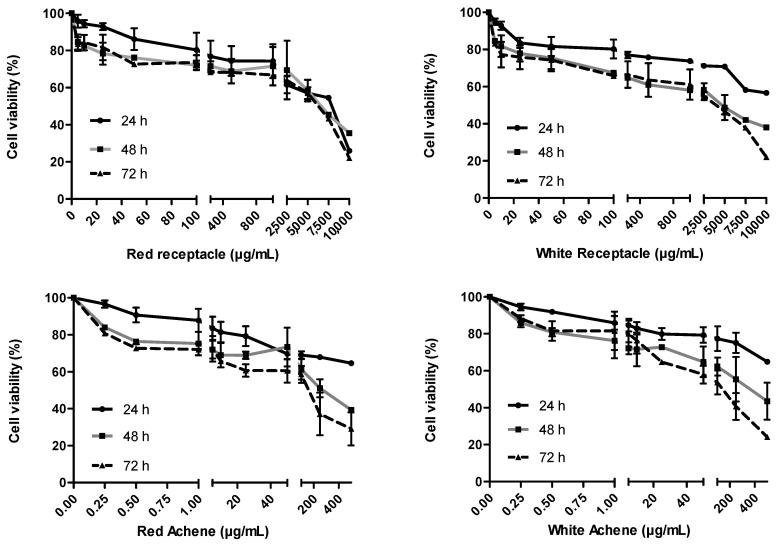
Viability of HepG2 cells determined by the MTT assay after incubation with different concentrations of red and white strawberry fruits (receptacles: 0–10,000 µg/mL and achenes: 0–400 µg/mL) during 24, 48, and 72 h.

**Figure 3 foods-13-00110-f003:**
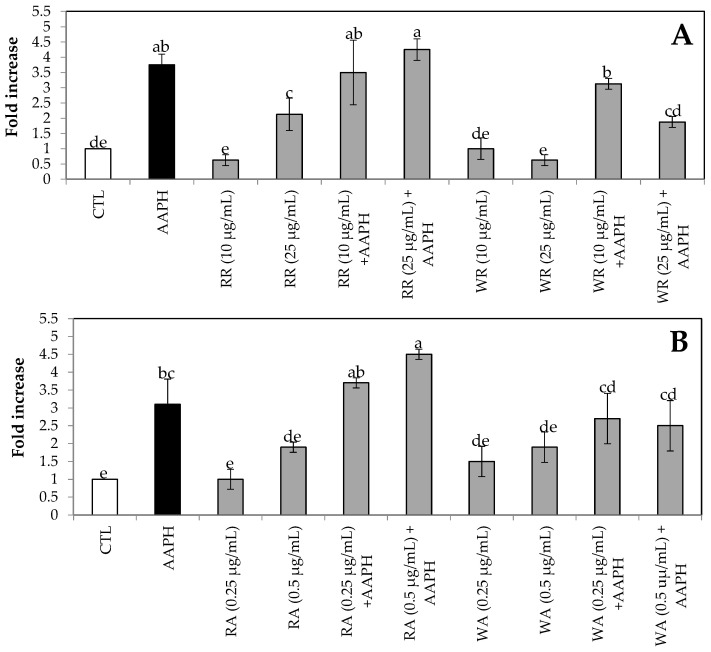
Intracellular reactive oxygen species (ROS) accumulation in HepG2 cells. Cells were preincubated with the indicated fruit extract ((**A**) receptacle and (**B**) achenes) at two concentrations (10 and 25 µg/mL for receptacle and 0.25 and 0.5 µg/mL for achenes) and/or then stressed with AAPH for 24 h. Values are expressed as the mean ± SE of three independent experiments (n = 3). Columns belonging to the same set of data with different superscript letters are significantly different (*p* < 0.05). CTL: cells with-out treatment; AAPH: cells incubated with AAPH; RR: cells incubated with red receptacle; WR: cells incubated with white receptacle; RR + AAPH: cells preincubated with red receptacle, then stressed with AAPH; WR + AAPH: cells preincubated with white receptacle, then stressed with AAPH; RA: cells incubated with red achenes; WA: cells incubated with white achenes; RA + AAPH: cells preincubated with red achenes, then stressed with AAPH; WA + AAPH: cells preincubated with white achenes, then stressed with AAPH.

**Figure 4 foods-13-00110-f004:**
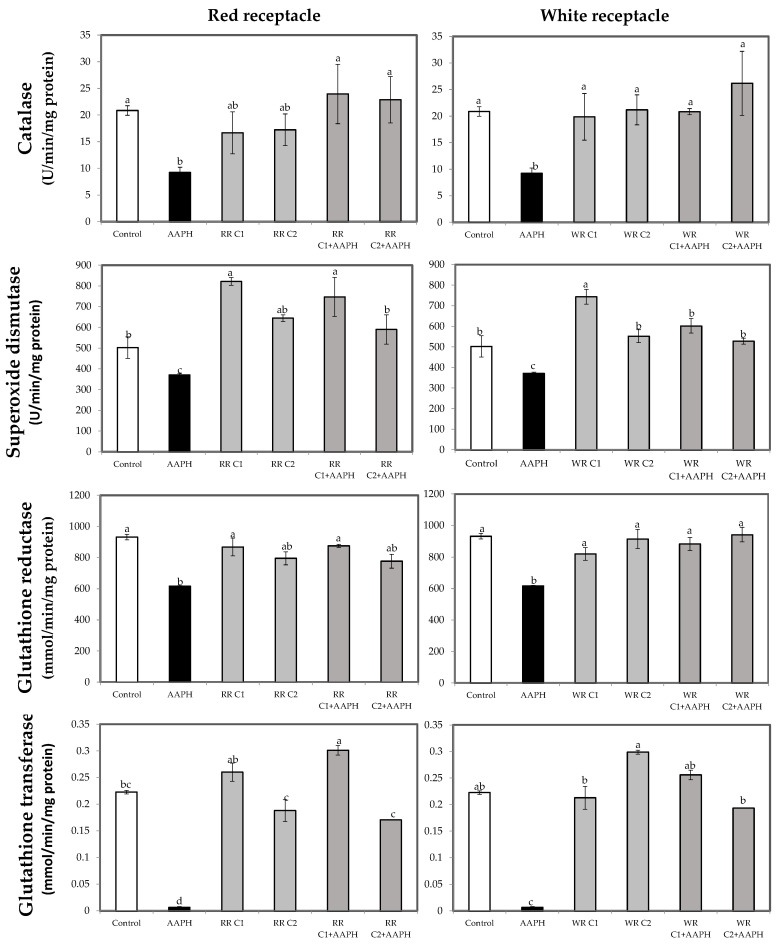
Catalase (CAT; U/min/mg protein), superoxide dismutase (SOD; U/min/mg protein), glutathione reductase (GR; mmol/min/mg protein), and glutathione transferase (GST; mmol/min/mg protein) activities in HepG2 cells treated with two concentrations of receptacle, red or white, for 24 h or AAPH (2.5 mM) for 24 h and two concentrations of receptacle, red or white, then stressed with AAPH. Data are expressed as mean values ± SE. Columns with different superscript letters are significantly different (*p* < 0.05). CTL: cells without treatment; AAPH: cells incubated with AAPH; RR: cells incubated with red receptacle; WR: cells incubated with white receptacle; RR + AAPH: cells preincubated with red receptacle, then stressed with AAPH; WR + AAPH: cells preincubated with white receptacle, then stressed with AAPH; C1: concentration 1 (10 µg/mL); C2: concentration 2 (25 µg/mL).

**Figure 5 foods-13-00110-f005:**
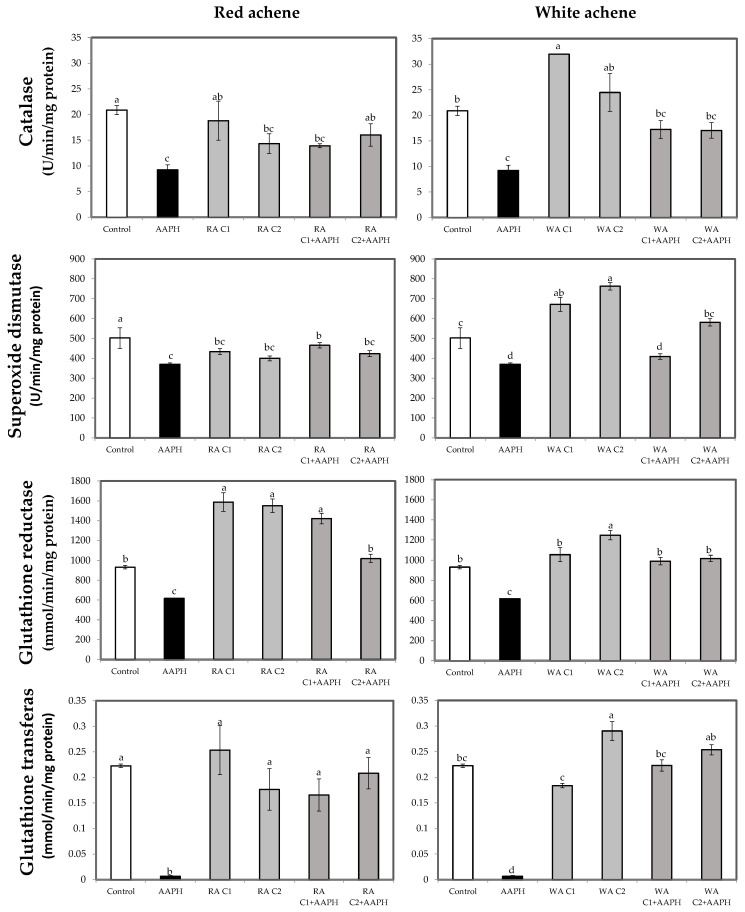
Catalase (CAT; U/min/mg protein), superoxide dismutase (SOD; U/min/mg protein), glutathione reductase (GR; mmol/min/mg protein), and glutathione transferase (GST; mmol/min/mg protein) activities in HepG2 cells treated with two concentrations of achenes, from red or white fruits, for 24 h or AAPH (2.5 mM) for 24 h and two concentrations of achenes, from red or white fruits, then stressed with AAPH. Data are expressed as mean values ± SE. Columns with different superscript letters are significantly different (*p* < 0.05). CTL: cells without treatment; AAPH: cells incubated with AAPH; RA: cells incubated with red achenes; WA: cells incubated with white achenes; RA + AAPH: cells preincubated with red achenes, then stressed with AAPH; WA + AAPH: cells preincubated with white achenes, then stressed with AAPH; C1: concentration 1 (0.25 µg/mL); C2: concentration 2 (0.5 µg/mL).

**Figure 6 foods-13-00110-f006:**
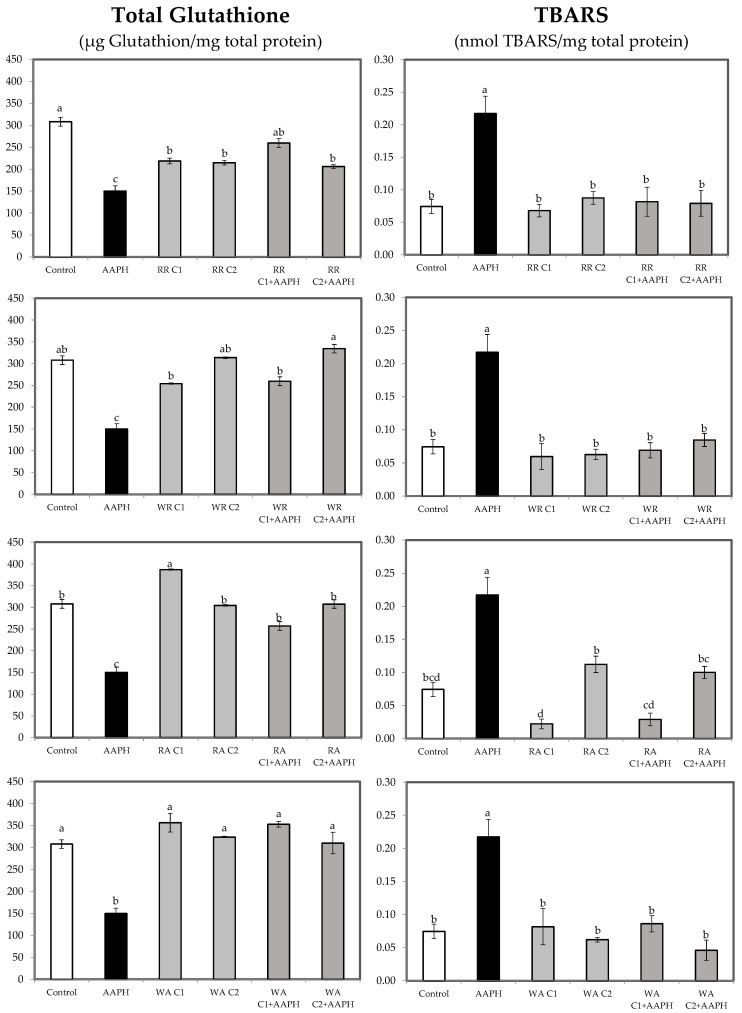
Total glutathione and TBARS in HepG2 cells treated with two concentrations of receptacle and achenes for 24 h or AAPH (2.5 mM) for 24 h with two concentrations of receptacle and achenes, then stressed with AAPH. Data are expressed as mean values ± SE. Columns with different superscript letters are significantly different (*p* < 0.05). CTL: cells without treatment; AAPH: cells incubated with AAPH; RR: cells incubated with red receptacle; WR: cells incubated with white receptacle; RR + AAPH: cells preincubated with red receptacle, then stressed with AAPH; WR + AAPH: cells preincubated with white receptacle, then stressed with AAPH; RA: cells incubated with red achenes; WA: cells incubated with white achenes; RA + AAPH: cells preincubated with red achenes, then stressed with AAPH; WA + AAPH: cells preincubated with white achenes, then stressed with AAPH; C1: concentration 1 (10 µg/mL for receptacle and 0.25 µg/mL for achenes); C2: concentration 2 (25 µg/mL for receptacle and 0.5 µg/mL for achenes).

**Table 1 foods-13-00110-t001:** Total phenolic content, total flavonoid content, total anthocyanin content, total tannin acid content, and antioxidant capacity of raw fruits from white and red mature strawberry fruits in both receptacle and achenes.

	Total Phenolic Content				Total Flavonoids Content				Total Anthocyanin Content				Total Tannin Content				Antioxidant Capacity			
	(mg GAE/100 g FW)				(mg CAE/100 g FW)				(mg PE/100 g FW)				(mg TAE/100 g FW)				(µmol TE/100 g FW)			
Red receptacle	166.82	±	5.16	A	41.65	±	1.57	B	21.87	±	0.89	A	77.02	±	2.00	A	907.95	±	39.10	B
Red Achenes	5110.47	±	174.66	a	748.67	±	30.26	a	13.65	±	0.66	a	2952.30	±	126.25	b	18886.28	±	770.14	a
	(35.77; 17.66%)				(5.24; 11.18%)				(0.10; 0.44%)				(20.67; 21.16%)				(132.20; 12.71%)			
White receptacle	154.80	±	5.48	B	49.60	±	1.03	A	<LOQ			B	75.09	±	2.06	A	1190.17	±	47.46	A
White achenes	5416.38	±	80.23	a	544.61	±	10.84	b	<LOQ			b	3212.79	±	90.16	a	18969.09	±	584.56	a
	(37.91; 19.67%)				(3.81; 7.14%)								(22.49; 23.05%)				(132.78; 10.04%)			

Different letters indicate significant differences for each parameter (*p* < 0.05); upper-cases and lower-cases indicate significant differences between receptacles and between achenes, respectively. Figures in brackets represent the values and percentage of each parameter relative to the total achenes per fruit (achenes represent 0.7% of total fruit weights), according to [10]. Units: mg standard or µmol equivalent per 100 g of fresh weight (FW). LOQ: limit of quantification.

**Table 2 foods-13-00110-t002:** Composition of phenolic acids and hydrolysable tannins, flavanols, flavonols, and anthocyanins in receptacle and achenes of red strawberry (*Fragaria x ananassa*, Duch, cv. Fortuna) and white strawberry (*Fragaria vesca* spp. XXVIII), determined by HPLC–DAD. All analyses were performed in triplicate. Data (mean ± SE) are expressed in mg/100 g of fresh weight (FW).

		Red Receptacle				Red Achene				White Receptacle				White Achene			
		(mg/100 g FW)				(mg/100 g FW)				(mg/100 g FW)				(mg/100 g FW)			
Phenolic acids and hydrolisable tannins	Gallic acid	1.49	±	0.25	A	<LOQ				0.68	±	0.07	B	<LOQ			
	Chlorogenic acid	17.45	±	2.80	A	<LOQ				<LOQ			B	<LOQ			
	Caffeic acid	0.81	±	0.05	A	<LOQ				<LOQ			B	<LOQ			
	p-Coumaric acid	0.50	±	0.02	A	<LOQ				<LOQ			B	<LOQ			
	Ellagic acid	229.72	±	11.49	A	1293.25	±	92.74	a	77.43	±	7.57	B	601.27	±	23.88	b
	trans-Cinamic acid	0.42	±	0.02	A	<LOQ				<LOQ			B	<LOQ			
	Total	250.39				1293.25				78.11				601.27			
Flavanols	Procyanindin B1	32.20	±	4.30	A	<LOQ				7.99	±	1.14	B	<LOQ			
	Catechin	5.58	±	0.31	B	<LOQ				12.77	±	0.59	A	<LOQ			
	Procyanidin B2	9.49	±	0.43	A	<LOQ				<LOQ			B	<LOQ			
	Total	47.27				<LOQ				20.77				<LOQ			
Flavonols	Chlorogenic acid	17.45	±	2.80	A	<LOQ				<LOQ			B	<LOQ			
	Rutin	1.59	±	0.36	A	11.47	±	1.22	a	<LOQ			B	1.27	±	0.36	b
	Myricetin	0.79	±	0.07	A	3.75	±	0.07	b	0.56	±	0.02	A	48.34	±	2.74	a
	Quercetin	<LOQ				<LOQ				<LOQ				<LOQ			
	Kaempferol	<LOQ				<LOQ				<LOQ				<LOQ			
	Total	2.37				15.22				0.56				49.62			
Anthocyanins	Cya-3-glc	<LOQ				2.43	±	0.93	a	<LOQ				<LOQ			b
	Pel-3-glc	19.29	±	1.47	A	4.42	±	1.08	a	<LOQ			B	<LOQ			b
	Pel-3-rut	0.46	±	0.03	A	<LOQ				<LOQ			B	<LOQ			
	Peo-3-glc	<LOQ				<LOQ				<LOQ				<LOQ			
	Total	19.75				6.84				<LOQ				<LOQ			

Different letters indicate significant differences for each antioxidant group (*p* < 0.05); upper-cases are significant differences between receptacles and lower-cases are significant differences between achenes. LOQ: limit of quantification.

## Data Availability

The datasets used and/or analyzed during the current study are available from the corresponding author upon reasonable request.

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
