# Peer review of "Relation between Strawberry Fruit Redness and Bioactivity: Deciphering the Role of Anthocyanins as Health Promoting Compounds"

_foods, 2023, doi:10.3390/foods13010110_

Round 1

Reviewer 1 Report

Comments and Suggestions for Authors

The scientific work entitled : Relation between strawberry fruit redness and bioactivity: deciphering the role of anthocyanins as health promoting compounds, is well presented but there are some remarks that need to be made:

The title should be reworded to include the basic plant of this research.

Abstract: should consist of the results in order to attract scientific researchers.

Have the authors identified this plant? If so, this should be added.

A list of products and a list of abbreviations should be added.

Figure 1: must be in a clear format.

Throughout the manuscript, the units of the results should be checked.

Conclusion: perspectives should be added.

Comments on the Quality of English Language

The manuscript needs some minor language corrections.

Author Response

Comments and Suggestions for Authors

-The scientific work entitled:  Relation between strawberry fruit redness and bioactivity: deciphering the role of anthocyanins as health promoting compounds, is well presented but there are some remarks that need to be made:

We thank the referee for his positive comments on our manuscript.

-The title should be reworded to include the basic plant of this research.

We are not sure what the reviewer means. The title already includes the generic plant name of the species used in this research. Species are already indicated in the material and methods section, but we have also included them in the abstract.

-Abstract: should consist of the results in order to attract scientific researchers.

We have included the scientific name of the species used in our research in the abstract and a sentence to underline the main result of our research.

-Have the authors identified this plant? If so, this should be added.

We do not understand what the reviewer means. If he/she means the name of the variety of the F. vesca spp., we have included the accession number (XXVIII) in the Malaga germplasm bank.

-A list of products and a list of abbreviations should be added.

Done as suggested by the reviewers at the end of the manuscript.

-Figure 1: must be in a clear format.

Done as suggested by the reviewers.

-Throughout the manuscript, the units of the results should be checked.

Done as suggested by the reviewers.

-Conclusion: perspectives should be added.

Lines 518-521 are the perspectives that emerged from our study.

Reviewer 2 Report

Comments and Suggestions for Authors

The manuscript  entitled “Relation between strawberry fruit redness and bioactivity: deciphering the role of anthocyanins as health promoting compounds have showed that white receptacle extracts (deprived of anthocyanins) were able to protect cells from oxidative damage to a greater extent than red fruits. The manuscript describes polyphenols, e.g. spectrophotometric and HPLC techniques were used for characterizing the content of phenolic compounds, including anthocyanins, in fruits of red and white strawberry species (distinguishing receptacle from achene). In addition, the effect of these extracts on the reduction of intracellular ROS in HepG2 cells was tested, as well as on the activity of antioxidant enzymes (SOD, CAT, GR GTs, GSH, GSSG)  and the quantification of cell oxidation markers.

The manuscript was written carefully and well in terms of language. It should be noted that the research was properly planned and discussed.

Minor issues:

Line 100-108: please describe methodology for each analyses e.g. TPC, TFC, TTC, AC and TEAC (how to prepare samples, which was the control? and how measurements and calculated each phytochemical contents?

Line 169-176 please describe methodology and citation (how to prepare samples, which was the control? and how measurements and calculated each enzyme activities?

Figure 1: change axis descriptions, on the figure 1 is “ Dim 1 24.12% and Dim 2 65.50%’, but in the text is “the first PC 319 explained 24.12% of the total variance and the second PC explained 65.50%” (Line 319-320)

Author Response

Comments and Suggestions for Authors

- The manuscript entitled “Relation between strawberry fruit redness and bioactivity: deciphering the role of anthocyanins as health promoting compounds” have showed that white receptacle extracts (deprived of anthocyanins) were able to protect cells from oxidative damage to a greater extent than red fruits. The manuscript describes polyphenols, e.g. spectrophotometric and HPLC techniques were used for characterizing the content of phenolic compounds, including anthocyanins, in fruits of red and white strawberry species (distinguishing receptacle from achene). In addition, the effect of these extracts on the reduction of intracellular ROS in HepG2 cells was tested, as well as on the activity of antioxidant enzymes (SOD, CAT, GR GTs, GSH, GSSG) and the quantification of cell oxidation markers.

The manuscript was written carefully and well in terms of language. It should be noted that the research was properly planned and discussed.

We thank the referee for his positive comments on our manuscript.

Minor issues:

- Line 100-108: please describe methodology for each analyses e.g. TPC, TFC, TTC, AC and TEAC (how to prepare samples, which was the control? and how measurements and calculated each phytochemical contents?

Done as suggested by the reviewers.

- Line 169-176 please describe methodology and citation (how to prepare samples, which was the control? and how measurements and calculated each enzyme activities?

Done as suggested by the reviewers.

- Figure 1: change axis descriptions, on the figure 1 is “Dim 1 24.12% and Dim 2 65.50%’, but in the text is “the first PC 319 explained 24.12% of the total variance and the second PC explained 65.50%” (Line 319-320)

Done as suggested by the reviewers.

Reviewer 3 Report

Comments and Suggestions for Authors

The manuscript, according its title, examines the realation between strawberry redness and bioactivity. First of all, redness as organoleptical quality criterion was not determined or assessed; it was only and mostly related to the presence of anthocyanins. The title of the manuscript is not relevant to its content.

Spectrophotometric assays, such as total phenolic content (TPC), total flavonoid content (TFC), total anthocyanin content (TAC), total tannic acid content (TTC) as well as antioxidant capacity (AC) by Trolox Equivalent Antioxidant Capacity (TEAC) assay were determined, according to previous publication of the same authors. TEAC assay is widely performed to assess the antioxidant capacity of an antioxidant, specifically the amount of free radicals that can be scavenged. A major problem associated with the TEAC assay is that the reaction between most antioxidants and the ABTS· radical is not completed within the time period applicable, resulting in underestimation of the TEAC value of these antioxidants, as well as the dependence of TEAC value on the concentration of the antioxidant. 

Finally, in order to estimate and elucidate the role of the anthocyanins and/or other bioactive compounds on strawberry's bioactivity, a more robust technique, such as LC-MS, for evaluating the phytochemical profile, would be appropriate.

Detailed comments:

L.36: antioxidants

L.97: type of homogenizer? Ultrasonic extraction could be better for the recovery.

L.103-105: The spectrophotometric assays could be described shortly. Additionally, authors should provide information details on the standard compounds in a separate paragraph.

L.112: analysed

L.189: Replace with "Spectrophotometric assays ...

L.231-232: Authors should explain the statement.

Table 2: How did authors presume that LOQ is statistically different than the relative content of red anchenes (4.42±1.08), since  LOQ was not determined?

Comments on the Quality of English Language

Minor editing is required.

Author Response

Comments and Suggestions for Authors

-The manuscript, according its title, examines the relation between strawberry redness and bioactivity. First of all, redness as organoleptical quality criterion was not determined or assessed; it was only and mostly related to the presence of anthocyanins. The title of the manuscript is not relevant to its content.

Redness has been widely associated in most studies with anthocyanin content (Khoo et al., 2017; Dzhanfezova et al., 2020) and in our study redness was assessed visually, as one of the two strawberry species studied is completely white, so redness cannot be determined. Moreover, the anthocyanin content on each type of fruits (non- detectable in white fruits) reinforces their close relation with the red color. For clarity, we have included a photo of the fruits of these two strawberries used in this study.

Considering this, we think the title is appropriate.

Khoo HE,  Azlan A, Tang ST, and Lim SM. 2017. Anthocyanidins and anthocyanins: colored pigments as food, pharmaceutical ingredients, and the potential health benefits. Food Nutr Res. 61(1): 1361779. doi: 10.1080/16546628.2017.1361779

Dzhanfezova T, Barba-Espín G, Müller R, Joernsgaard B, Hegelund JN, Madsen B, Larsen DH, Martínez Vega M and Toldam-Andersen TB. 2020. Anthocyanin profile, antioxidant activity and total phenolic content of a strawberry (Fragaria × ananassa Duch) genetic resource collection. Food Bioscience, 36,100620. https://doi.org/10.1016/j.fbio.2020.100620.

-Spectrophotometric assays, such as total phenolic content (TPC), total flavonoid content (TFC), total anthocyanin content (TAC), total tannic acid content (TTC) as well as antioxidant capacity (AC) by Trolox Equivalent Antioxidant Capacity (TEAC) assay were determined, according to previous publication of the same authors. TEAC assay is widely performed to assess the antioxidant capacity of an antioxidant, specifically the amount of free radicals that can be scavenged. A major problem associated with the TEAC assay is that the reaction between most antioxidants and the ABTS·radical is not completed within the time period applicable, resulting in underestimation of the TEAC value of these antioxidants, as well as the dependence of TEAC value on the concentration of the antioxidant.

We performed the TEAC assay as described in many other studies, and it is a widely accepted method by the scientific community in the way we have performed the assay on strawberry. As the reviewer suggests, perhaps it may have some criticism on the representativeness of the total antioxidant capacity (in fact, there are other complementary methods such as FRAP and DPPH that we already checked in a previous work Ariza et al. 2013) and on the time in which the reaction was performed. However, the time interval used is within the range proposed for the ABTS assay by Dong et al. 2015 (doi:10.1177/1934578X1501001239). In case the reaction time would have been insufficient to oxidize all reagents, this would not invalidate the comparison of the results between the two types of fruits, although there may be room for underestimation.

Dong J-W, Cai L, Xing Y, Yu J and Ding Z-T. 2015. Re-evaluation of ABTSï‚–+ Assay for Total Antioxidant Capacity of Natural Products. Natural Product Communications Vol. 10 (12).

-Finally, in order to estimate and elucidate the role of the anthocyanins and/or other bioactive compounds on strawberry's bioactivity, a more robust technique, such as LC-MS, for evaluating the phytochemical profile, would be appropriate.

The reviewer is probably right that the use of LC-MS would be more accurate than HPLC, but the technique used in our study allows us to obtain relevant differences between the profiles of the main antioxidant compounds described in strawberries.

Detailed comments:

-L.36: antioxidants

Text has been amended as suggested by the reviewer.

-L.97: type of homogenizer? Ultrasonic extraction could be better for the recovery.

We agree that the ultrasonic extraction could be better for the recovery but in our study an Orbital shaker (Unimax 1010, Heidolph) was used for homogenization.

-L.103-105: The spectrophotometric assays could be described shortly. Additionally, authors should provide information details on the standard compounds in a separate paragraph.

Text has been amended as suggested by the reviewer.

-L.112: analysed

Text has been amended as suggested by the reviewer.

-L.189: Replace with "Spectrophotometric assays ...

Text has been amended as suggested by the reviewer.

-L.231-232: Authors should explain the statement.

The authors have rewritten the sentence for better understanding.

-Table 2: How did authors presume that LOQ is statistically different than the relative content of red achenes (4.42±1.08), since LOQ was not determined?

Values under the limit of quantification (LOQ) are considered as 0 so the statistics has been performed with these values.
